# Australian hospital paediatricians and nurses' perspectives and practices for influenza vaccine delivery in children with medical comorbidities

Daniel A. Norman[1,2]*, Margie Danchin[3,4,5], Christopher C. Blyth[1,2,6,7], Pamela Palasanthiran[8,9], David Tran[10], Kristine K. Macartney[11,12,13], Ushma Wadia[1,14], Hannah C. Moore[1], Holly Seale[15]

1 Wesfarmers Centre of Vaccines and Infectious Diseases, Telethon Kids Institute, University of Western Australia, Crawley, Western, Australia, 2 School of Medicine, University of Western Australia, Crawley, Western, Australia, 3 Department of General Medicine, The Royal Children's Hospital, Parkville, Victoria, Australia, 4 Vaccine Uptake Group, Murdoch Children's Research Institute, Parkville, Victoria, Australia, 5 Department of Paediatrics, University of Melbourne, Melbourne, Victoria, Australia, 6 Department of Infectious Diseases, Perth Children's Hospital, Nedlands, Western, Australia, 7 Department of Microbiology, PathWest Laboratory Medicine, Nedlands, Western, Australia, 8 School of Women's and Children's Health, University of New South Wales, Sydney, New South Wales, Australia, 9 Infectious Diseases Service, Sydney Children's Hospitals Network, Randwick, New South Wales, Australia, 10 Department of Paediatrics, Northern Health, Epping, Victoria, Australia, 11 National Centre for Immunisation Research and Surveillance, University of Sydney, Sydney, New South Wales, Australia, 12 Department of Infectious Diseases and Microbiology, Children's Hospital Westmead, Westmead, New South Wales, Australia, 13 School of Paediatrics and Child Health, University of Sydney, Sydney, New South Wales, Australia, 14 Department of Paediatrics, Fiona Stanley Hospital, Murdoch, Western, Australia, 15 School of Population Health, University of New South Wales, Sydney, New South Wales, Australia

* daniel.norman@telethonkids.org.au

**Data Availability Statement:** Data cannot be shared directly publicly because for the ethics requirements of this study to maintain anonymity

## Abstract

### Introduction

Influenza vaccination of children with medical comorbidities is critical due their increased risks for severe influenza disease. In Australia, hospitals are an avenue for influenza vaccine delivery to children with comorbidities but are not always effectively utilised. Qualitative enquiry sought to ascertainment the barriers and enablers for influenza vaccination recommendation, delivery, and recording of these children at Australian hospitals.

### Methods

Semi-structured interviews and discussion group sessions were conducted with paediatricians and nurses at four tertiary paediatric specialist hospitals and two general community hospitals in three Australian states. Transcripts from interviews and group sessions were inductively analysed for themes. The Capability, Opportunity, Motivation, and Behaviour (COM-B) model was used to explore the elements of each theme and identify potential interventions to increase influenza vaccination recommendation and delivery behaviours by providers.

of the participants as the data contains potentially identifying information of participants. Data are available from the Ethics Committee (contact via Child and Adolescent Health Services of the Depart of Health of the Government of Western Australia [Project Registration Number: RGS0000001382]) for researchers and applicants who meet the criteria for access to confidential data. Individual application for access to this data can be done through the Child and Adolescent Health Services of the Depart of Health of the Government of Western Australia (Project Registration Number: RGS0000001382). For a non-author contact for data availability please contact Ethics and Governance of the Child and Adolescent Health Service, Western Australia Health: CAHS. ethics@health.wa.gov.au.

**Funding:** PP provided funding for DN's airline travel and accommodation during discussion groups and interviews conducted at Children's Hospital at Westmead and Sydney Children's Hospital, Randwick through PP's Funding from the Department of Immunology and Infectious Diseases, Sydney Children's Hospitals Network, Randwick. DN's airline travel and accommodation during discussions and interviews conducted at The Royal Children's Hospital, Melbourne and Northern Hospital, Epping was provided through his University of Western Australia's HDR Funding: 20761103. No other funding for this project was sought or provided. The funders had no role in the study design, data collection and analysis, decision to publish, or preparation of the manuscript.

**Competing interests:** I have read the journal's policy and the authors of this manuscript have the following competing interests: A/Prof Seale has previously received funding from vaccine manufactures for investigator driven research and for presenting at workshops. This funding was not associated with this research and this does not alter our adherence to PLOS ONE policies on sharing data and materials. All other authors have no competing interests to declare. All authors have approved submission of the manuscript for publication to PLOS ONE.

## Results

Fifteen discussion sessions with 28 paediatricians and 26 nurses, and nine in-depth interviews (five paediatricians and four nurses) were conducted. Two central thematic domains were identified: 1. *The interaction between hospital staff and parents/patients for influenza vaccine recommendation*, and 2. *Vaccination delivery and recording in the hospital environment*. Six themes across these domains emerged detailing the importance of dedicated immunisation services, hospital leadership, paediatricians' vaccine recommendation role, the impact of comorbidities, vaccination recording, and cocooning vaccinations. Supportive hospital leadership, engaged providers, and dedicated immunisation services were identified as essential for influenza vaccination of children with comorbidities in Australian hospital.

## Conclusion

Recommendation of influenza vaccination for Australian children with comorbidities is impacted by the beliefs of paediatricians and the perceived impact of influenza disease on children's comorbidities. Dedicated immunisation services and supportive hospital leadership were drivers for influenza vaccine delivery at hospitals. Future interventions targeting hospital-based influenza vaccine delivery for children with comorbidities should take a rounded approach targeting providers' attitudes, the hospital environment and leadership support.

## Introduction

Children with medical comorbidities are more likely to have severe disease outcomes and require intensive treatment from influenza disease compared to previously healthy children [1]. Accordingly, influenza vaccination is funded by Australia's National Immunisation Program (NIP) for children 6-month-old and older with medical comorbidities. These comorbidities are detailed in the Australian Immunisation Handbook and are defined by their association with increased risk for severe influenza disease including chronic respiratory, cardiac, neurological, genetic, immunosuppression, malignancies, haematological, and metabolic conditions [2] Despite this, influenza vaccine uptake in Australian children remains inadequate with past evaluations showing less than 40% coverage [3].

In Australia, the responsibility for childhood immunisations falls predominantly to primary care and community immunisation services [4]. However, a recent survey of families attending Australian paediatric outpatient clinics identified the hospital environment and hospital clinical staff are critical for influenza vaccine delivery to children with medical comorbidities [5]. Children with comorbidities in Australia regularly attend hospital outpatient clinics or large paediatrics clinics linked with hospitals for their clinical treatment and management. Their care is usually overseen by subspeciality physicians/paediatricians or general paediatricians with specific skills in managing specific comorbidity groups [6]. Our previous work identified that a recommendation by the child's hospital physician/paediatrician was a stronger predictor for reported influenza vaccine uptake compared to general practitioners, nurses, and family members. Additionally, 80% of survey respondents were willing for their child to receive influenza vaccination during future hospitalisations [5].

However, ensuring a recommendation and facilitating influenza vaccine delivery to patients remains a challenge. A previous Australian study examining the factors impacting promotion

and delivery of immunisation at a paediatric hospital identified multiple barriers including the hospital's limited capacity for influenza vaccine delivery, competing clinical priorities, and a lack of awareness for vaccine recommendations. Paediatricians and nurses with experience in treating children with influenza strongly prioritised influenza vaccine recommendation for their patients with comorbidities. However, this study was of a single tertiary paediatric site limiting generalisability [7].

This study aimed to understand the factors that impact recommendation and delivery of influenza vaccination to children with comorbidities at Australian hospitals with and without dedicated immunisation services. Dedicated immunisation services are specialised clinical teams and services located within multiple Australian paediatric hospitals that facilitate immunisations and related services to inpatients and outpatients, often through an opportunistic model for both childhood schedule and non-schedule immunisations including influenza vaccines [8]. Additionally, dedicated immunisation services act as resources hubs for immunisation-related education and counselling to patients and their families as well as other clinical teams [8]. We sought to explore perspectives of paediatricians and nurses for influenza vaccine recommendation and delivery working at Australian paediatric tertiary and major community hospitals. These insights are needed to inform future intervention design and implementation to increase influenza vaccine uptake for children with comorbidities.

## Methods

Qualitative in-depth interviews and discussion groups sessions with hospital staff from four tertiary paediatric hospitals and two metropolitan general hospitals were conducted between May 2019 to February 2020 in Western Australia, New South Wales, and Victoria, Australia. These hospitals were Perth Childrens Hospital, Fiona Stanley Hospital, the Royal Children's Hospital Melbourne, Northern Hospital Epping, Children's Hospital at Westmead, and Sydney Children's Hospital, Randwick. Hospitals were selected to include those with and without dedicated immunisation services or comparable onsite vaccination services. Purposeful recruiting of hospital staff was facilitated through direct contact with key clinical leaders of subspeciality departments and hospital immunisations teams at each site. Researchers coordinated with department and team leaders to identify suitable paediatricians and nurses from their respective departments for participation. Hospital-based subspecialist and general paediatricians as well as nurses who were involved in recommendation and delivery of influenza vaccines to patients or planning of departments' influenza immunisation programs were targeted for participation. Ethics approval was received from the Child and Adolescent Health Services at the Western Australian Department of Health (Project Number: RGS0000001382) with respective clinical governance approval from each site. Informed written consent was obtained from all participating paediatricians and nurses.

Fifteen discussion sessions and nine in-depth interviews were conducted in a semi-structured format taking between 25 and 50 minutes by DN, audio recordings were created with paediatricians and nurses' permission. A pre-established interview guide based on previous studies' results [5, 7] and clinical guidance was used to guide open-ended discussion (S1 Appendix). Interview question topics included paediatricians and nurses' knowledge, beliefs, and practices regarding influenza infection, and vaccination for children with medical comorbidities as well as structural and temporal barriers impeding influenza vaccine recommendation and delivery. Theoretical patient models were used to facilitate discussion around comorbidity related risks for influenza disease and vaccine recommendations. Reactive impromptu questions were used to continue the discussion as well as explore different ideas and themes of discussion topics. Member checking was used to clarify responses by asking for

participants' validation of summaries of what was spoken about at each session's or interview's conclusion. Most discussion sessions and interviews were conducted at hospital sites face-to-face by DN, with a minority conducted by telephone with all audio recordings transcribed verbatim. There were no relevant relationships between the interviewer and interviewees prior to conducting of interviews.

The step-wise process developed by *Braun and Clarke* [9] for thematic analysis and coding of audio transcriptions was used, with analysis conducted in *NVivo® 12* [10] detailed by 1) data familiarisation, 2) generation of initial codes, 3) searching for themes, 4) reviewing of themes, 5) defining of themes and 6) write-up of analysis. Analysis began concurrently with the completion of the first interview. An inductive approach was used during the data analysis to develop codes linked to the data. A pre-determined coding handbook or theme framework was not used, instead the process used open coding to begin with, followed by a coding book once consensus had been reached on the codes. Themes were formed in-situ from participants' perceptions, statements, and sentiments. DN conducted coding and thematic analysis with HS as a secondary coder. MD reviewed the codes and themes that arose from the analysis by DN and HS. Feedback was not sought from discussions session and interviewees' participants due to the increased time burden.

Themes arising from transcripts were used to build conceptual thematic domains of the social and physical environments of Australian hospitals that influence influenza vaccination of children with comorbidities. These domains with their respective themes were then evaluated with the *Capability*, *Opportunity*, *Motivation*, *and Behaviour* model (COM-B) to understand the elements within each theme that influence the behaviours of providers for recommending and delivering influenza vaccination to children with comorbidities [11]. COM-B has previously been utilised by one of the authors (MD) for understanding the behaviours of Australian midwives for recommendation and delivery of prenatal and childhood vaccines [12]. The findings of this study informed the design of a multi-component antenatal intervention bundle of vaccine reminder prompts, provider communication training, and vaccine fact sheets [13].

## Results

Fifteen discussion sessions were undertaken across Western Australia, New South Wales, and Victoria at four specialist tertiary paediatric hospitals and two general hospitals admitting paediatric patients. In total 28 paediatricians and 26 nurses took part in group discussion sessions. The remaining nine in-depth one-on-one interviews were conducted with five paediatricians and four specialist nurses. The different number of paediatricians and nurses as well as their subspeciality from the tertiary paediatric hospitals and general hospitals are detailed in Table 1. All participating paediatricians were at a consultant or advance trainee level and nursing staff were at a registered nurse level or higher. Further details of participants are not disclosed to maintain anonymity.

Analysis of all transcripts gave rise to two central domains and six key themes. The two identified domains that were found to influence vaccine recommendation, delivery and recording: 1. *The interaction between paediatricians and nurses with parents/patient for influenza vaccine recommendation* and 2. *The hospital environment for influenza vaccination delivery and recording*. The first domain encompasses paediatricians and nurses' beliefs, perceptions, and actions that directly impact their interaction with parents/patient for influenza vaccine recommendation. These include paediatricians' perceptions of their responsibility for promoting vaccination recommendations, how a child's comorbidities impact influenza disease severity and necessitate protection through vaccination and need for family cocooning

**Table 1. Paediatricians and nurses participating in group discussions and in-depth interviews.**

| Hospital type | Subspeciality | Paediatricians (n) | Nurses (n) |
|---|---|---|---|
| **Tertiary Paediatric Hospitals** | Cardiology | 1 | 1 |
| | Diabetes, immunology and endocrinology | 3 | 5 |
| | General Paediatrics | 3 | 0 |
| | Respiratory medicine | 6 | 7 |
| | Infectious diseases and immunisation services | 5 | 4 |
| | Neurology | 3 | 2 |
| | Nephrology | 4 | 2 |
| | Oncology | 2 | 1 |
| **General Community Hospitals** | General paediatrics | 6 | 8 |
| | Immunisation services | 0 | 1 |

vaccinations. The second domain details how elements of hospital environments directly and indirectly impact vaccine delivery and recording. Support of immunisation resources and services by hospital leadership critically informed vaccination delivery with the hospital environment, whereas recording of vaccinations at hospitals were dictated by staffs' engagement with the Australian Immunisation Register. The domains, corresponding themes, COM-B components and potential interventions and intervention targets are outlined in Table 2.

## Paediatricians perceive their role is to be for influenza vaccine recommendation rather than direct vaccination of patients

Most paediatricians were strongly supportive of their role to recommend vaccination but were not necessarily supportive of their role in delivering influenza vaccination to patients with medical comorbidities. Paediatricians saw their primary responsibility to recommend vaccination but believed their capacity for vaccine delivery was greatly limited. "*I think as a doctor we say vaccinate and then pass it on to someone else to get it done, we are not chasing up every single patients' records*" [General Paediatrician]. Paediatricians expressed their clinical workload was

**Table 2. Domains, themes, COM-B components, interventions, and targets.**

| Thematic Domains | Themes | COM-B components | Potential interventions | Intervention targets |
|---|---|---|---|---|
| The interactions between paediatricians and nurses with parents/patients for influenza vaccine recommendation | Paediatricians perceive their role is to recommend rather than deliver vaccines. | Reflective motivation | Education and communication training | Paediatricians |
| | Providers perceive differences for influenza diseases risks and influenza vaccination necessity for different comorbidity types | Reflective motivation and automatic motivation | Education, role models (vaccine champions) | Paediatricians |
| | Clinicians support a cocooning vaccination strategy to protect their patients | Physical opportunity and physical capability | Process changes with cocooning vaccinations availability and cost | Patient families and the hospital environment |
| The hospital environment for influenza vaccination delivery and recording | Dedicated immunisation services are critical for supporting on-site influenza vaccination of patients | Physical opportunity and automatic motivation | Process changes, advocacy, and increased funding of dedicated immunisation services | The hospital environment |
| | Support and endorsement of influenza vaccination programs by hospital leadership is essential for the success of programs | Social opportunity and automatic motivation | Financial support, advocacy and active promotion | The hospital environment |
| | Recording influenza vaccination on the Australian Immunisation Register remains a challenge | Phycological capability and social opportunity | Process changes and dedicated information/guidelines for AIR usage | Paediatricians, nurses and the hospital environment |

a major limiting factor in their capacity to deliver vaccines. "*Honestly when I start an outpatient clinic I am just thinking about how I am going to survive the clinic* (with) *the amount of patients I am seeing. . .when they talk to me about it I would then say yes, go to the immunisation centre.*" [Paediatric Neurologist]. Reliance on pre-existing systems for delivery once a paediatrician had given a recommendation was additionally highlighted "*For me it's someone else, on the ward if we had nurse in charge who would check all the inpatients records and note who needed one and then got them vaccinated.*" [Paediatric Neurologist].

## Providers perceive differences for influenza diseases risks and influenza vaccination necessity for different comorbidity types

Perceptions around the risk of influenza disease for their patients was critical to informing the attitudes of paediatricians and nurses towards influenza vaccination and subsequently their likelihood of giving a recommendation. Paediatricians consistently identified that the influenza vaccination beliefs held by a clinician were strongly impacted by their respective subspecialty or clinical focus. "*I suspect the respiratory doctors are promoting it more than the non-respiratory are. As flu is mainly respiratory disease.*" [Immunisation Specialist Paediatrician]. Paediatricians' perception of influenza as primarily a respiratory disease aligned with their beliefs about which comorbidities increase risk for severe influenza disease and complications. Consequently, informing influenza vaccination beliefs and recommendation practices. "*It's the patient population, as neuromuscular (patients) have restrictive lung disease so if they get the flu they will deteriorate and that is well recognised. . . outside the neuro-muscular clinic it is ad hoc.*" [Paediatric Neurologist].

Certain subspecialties spoke about stratifying their patients' risks from influenza infections based on their clinical comorbidities. Some paediatricians perceived official recommendation guidelines to be too extensive and not accurately reflect patients' risk "*The first list are high risk ones we believe are critical to have the vaccine and the second list is where it is less absolutely recommended. We recognise that with the Australian immunisation handbook that the conditions are so broad that you could find a recommendation for anyone.*" [Paediatric Cardiologist]. Furthermore, immunisation clinic staff highlighted that some subspecialties had expressed misunderstandings about influenza vaccination risks for patients with certain comorbidities. "*Endocrinology don't believe influenza vaccination is that important for this group of patients and can even cause them to decompensate. So, we don't get a lot of those kids through to the clinic and we can only do what we can do when the family comes.*" [Immunisation Centre Lead].

Paediatricians and nurses expressed that their influenza vaccination practices were also influenced by a patients' medical comorbidities and current health status. Timing of vaccination was a key factor for patients undergoing chemotherapy "*with our more intensive phases (chemotherapy) we would recommend they have it at the start of the intensive phase block or in between intensive phase.*" [Paediatric Oncologist]. A minority of participants discussed influenza vaccination delays due to patients' current medical status or treatment "*For cardiac patients they've been nervous if you give them a vaccine shortly after they've had surgery. They worry that the vaccine can cause fever and then not knowing if the fever from the vaccine or from a complication in the surgery.*" [Immunisation Specialist Paediatrician].

## Clinicians support a cocooning vaccination strategy of families to additionally protect their children with comorbidities

Protective cocooning through influenza vaccination of immediate family members was described as an additional measure to protect patients with medical comorbidities from influenza infection "*one thing we do with high-risk children is recommend other family members,*

*other siblings is getting vaccinated so that they can cocoon the child. . . I think there is more barriers to that than getting the patients vaccinated."* [Respiratory Paediatrician]. Cocooning was said to have occurred both through hospital dedicated immunisation services *"Even here at the hospital they will immunize free of charge for family members associated with a child with a chronic health condition."* [Respiratory Nurse] as well as the GP clinics, *"sometimes they can do it here but other times when it's too busy they will say to do it with your GP and they will prioritise the child then the family".* [Oncology Nurse]. However, potential costs of cocooning vaccines were highlighted as barrier for some families *"Its $25 dollars for a parent to get it here, I know because I was enquiring for a CF patient's family which it would be a barrier for them due to financial constraints. . . I know on Friday during clinic one family said to me "I'm not going to get mine done here because its $25 dollars and my GP will do it for $16. So, that's a barrier"* [Respiratory Paediatrician].

## Dedicated immunisation services are critical for supporting on-site influenza vaccination of patients

One of the key issues that emerged was the variation in practice between hospitals that had dedicated immunisation services and those that did not. At hospitals with dedicated immunisation services, paediatricians and nurses reflected that they had adequate capacity to deliver vaccines and vaccine education, including influenza vaccination, easily and efficiently to patients and their families. Dedicated immunisation services provided additional direct support to other departments' subspecialty staff through vaccine education and resources: *"I feel that it's kind of revolutionized. . . the way we think about vaccination and the accessibility, and I think that's really made a lot of difference. And the clinical staff of the centre have been very forthcoming in terms of providing information they'd been able to communicate well with the parents in terms of the pros and cons of the immunisation."* [Paediatric Immunologist].

Paediatricians from hospital sites with dedicated immunisation services viewed them as the primary avenue for outpatient vaccine delivery. *"We can send patients attending our outpatient clinic or the department for another reason to the centre for a flu vaccination. We strongly promote it with our patient groups."* [Respiratory paediatricians]. Many identified that dedicated services presence allowed paediatricians to feel comfortable recommending patients receive influenza vaccination as delivery through the clinic was easily accessible for families. *"The barriers (vaccine recommendation) are zero, if they ask if they should get a flu vaccine we say yes, after you leave me go straight to the vaccination centre across from the café. . ."* [Paediatric Neurologist].

Conversely, participants from hospitals without dedicated immunisation services believed that they had the capacity to recommend vaccination but limited capacity for delivery. Consequently, they spoke of having to rely on external resources including other hospital immunisation clinics. *"We also share our very complex patients with the local children's hospital so I tell patients that they are able to get their vaccines at the drop-in immunisation service when they have their appointments at the children's hospital."* [General Paediatrician].

Limited capacity and opportunity for vaccine delivery and parental/patient motivation to engage with dedicated immunisation services were identified as barriers. The physical location of clinics was specifically highlighted to be important for patients. *"I think the location of our drop-in centre is a little bit of a problem. It's not so front and centre in the hospital. It's not on a high traffic area. So, I think that discourages patients walking to it."* [Immunisation Specialist Paediatrician]. Concerns of adequate resources to meet demands on dedicated immunisation services were also identified. *"The immunisation centre only has three rooms. . . But as you can see there is a line out the door and with the limit on space we are limited on nursing staff. So, the*

*executive is asking if we can do more, and I say; do more than 167 patients in a day? Not with the physical space we have now."* [Immunisation Centre Lead]. Despite these perceived barriers dedicated immunisation services were overwhelmingly seen to play a critical role in influenza vaccination for their respective hospitals' and a positive asset for subspecialty departments.

## Support and endorsement of influenza vaccination programs by hospital leadership is essential for the success of programs

Paediatricians and nurses directly involved in influenza vaccination program management consistently expressed the critical role hospital leadership have in successful vaccination programs. When hospital leadership was discussed by participants, they were perceived as potential advocates and financial supporters of hospital-based vaccination programs. Critically, it was recognised that the hospital leadership needed publicly endorse these programs within the hospital community and to clinical leadership through promotion and messaging. Their support allows for hospital-wide structural changes to occur and the establishment of dedicated immunisation services. Strong support and advocacy for programs by leadership allowed for easy implementation of programs within sites. "This issue (influenza vaccination) has been raised by the ombudsmen in the 2015 Child death report in New South Wales and tabled at the New South Wales parliament. So, I feel like the hospital executive has taken notice and is keen that we are particularly targeting children at high risk with vaccination." [Immunisation Specialist Paediatrician]. Leadership had also been utilised directly to communicate the importance of influenza vaccination to paediatricians and nurses. "We also send out all-user emails (written) by the hospital executive with the goal of clinicians recommending that, for example, that the 2019 flu vaccine is here and available. Here's how you recommend it to your kids." [Immunisation Specialist Paediatrician].

Immunisation staff primarily perceived the increase in patients' safety to be hospital leaderships' primary purpose for wanting to support improvements in influenza vaccination. "From an executive perspective, it was all about keeping our patients safe and most of the time that gets people across the line." [Infection Control Nurse]. However, if promotion or creation of vaccination programs did not align with or was counterproductive to leaderships' other goals, programs' implementation was not feasible. "I don't think it would be very positive due to the financial climate of the hospital executive, but if it was cost neutral or cost positive then they may consider it." [General Paediatrician]

## Recording and checking influenza vaccination records of patients on the Australian Immunisation Register is challenging

There was wide consensus on paediatricians' and nurses' ability to recommend and deliver influenza vaccination for their patients through the hospital space. However, participants expressed their capacity to record vaccine delivery in national or state-wide registries was very limited, as was their ability to easily check patients' vaccination status through the Australian Immunisation Registry (AIR) [14] *"It's actually very difficult to identify the patients we are seeing, who they are and what they have previously received."* [Renal Specialist Paediatrician].

Paediatricians and nurses' ability to check patients' influenza vaccination status was hampered by issues of record systems including: 1) awareness, *"But there isn't a way for us to record if the patient had the flu vax during the season rather than just asking them if they had it."* [Cardiology Nurse]; 2) capacity, *"There is only a limited number of staff who have access to AIR"* [Paediatric Nurse]; 3) accessibility, *"I think it should be recorded on AIR but I don't do it routinely because it is a hard process to login"* [General Paediatrician] and 4) functionality *"We're giving up to 700 vaccines a day. We did over 20,000 in total this year. So, and there was no*

*capacity to do like a spreadsheet data dump or anything. So, we didn't have the capacity"* [Paediatric Nurse]. Additionally, participants perceived efforts to manually check patients' vaccination status through clinical records or questioning patients were too cumbersome to warrant doing so. *"No, we don't actively go searching. I'm not sure. It is not a question that is absolutely asked at every clinic. Have you had your flu vaccine or not? I would say for some families, it would be. But I don't know that it's a routine question all the time."* [Neurology Nurse].

Issues outside the hospital space with records were also noted *"immunisation are on the child's page but there is not incentive for GPs to submit that data for the flu shot, it's not compulsory. . . we don't have such accurate information about children getting the flu vaccine outside of the hospital.* [Respiratory Paediatricians]. Hospital electronic medical records with influenza vaccination status details have been implemented at a limited number of Australian hospital sites. However, legal issues with interactions between internationally owned electronic medical records and national public health records were identified *"And the other thing that is a big barrier is the immunisation register it is fantastic that it's all of life but not linked to our electronic records. We tried but there were legal issues due to our system EPIC being from the US."* [Immunisation Specialist Paediatrician]. Despite the necessity of influenza vaccination records, awareness barriers, capacity, and legal issues strongly limited clinical staffs' ability and desire to record or check patients' vaccination status.

### Analysis of themes

From the six themes identified from interviews and discussion sessions, three interlinked components of a successful vaccination program were identified: 1) hospital leadership, 2) dedicated immunisation services, and 3) engaged hospital paediatricians and nurses. These elements interacted with each other to directly or indirectly influence influenza vaccination recommendation, delivery, and recording for children with comorbidities. Engaged and knowledgeable hospital paediatricians and nurses as well as dedicated services were seen to directly influence influenza vaccine uptake in children with comorbidities through vaccine recommendation and easy delivery respectively. Additionally, supportive hospital leadership was identified as a key factor for vaccine uptake in children through policies, funding, and intra-hospital promotion/directives.

## Discussion

This qualitative enquiry with hospital paediatricians and nurses illustrated variation in care and diverse perspectives across Australian hospitals for influenza vaccination of children with comorbidities. Commonalities in perspectives and experiences were also identified across hospital sites and Australian states. The flow of information, promotion, and advocacy for influenza vaccine recommendation and delivery were key elements in the hospital environment as described in Fig 1. Dedicated immunisation services, hospital leadership, and perceptions of comorbidity risk and the role of health care providers in vaccine recommendation were critical influences for paediatricians and nurses' capacity to recommend, deliver and record influenza vaccination.

Paediatricians identified that remembering the recommendations for influenza vaccination, beliefs around the impact of comorbidities on influenza disease severity, and their time capacity strongly influenced their behaviours. Participating paediatricians acknowledged that while influenza disease constituted a risk to children's health, they emphasized the tension between actively recommending influenza vaccine for their patients versus the perceived impact on their clinical workload. While acknowledging influenza vaccination recommendations were within their clinical remit, paediatricians saw direct vaccine delivery to not be part of their role

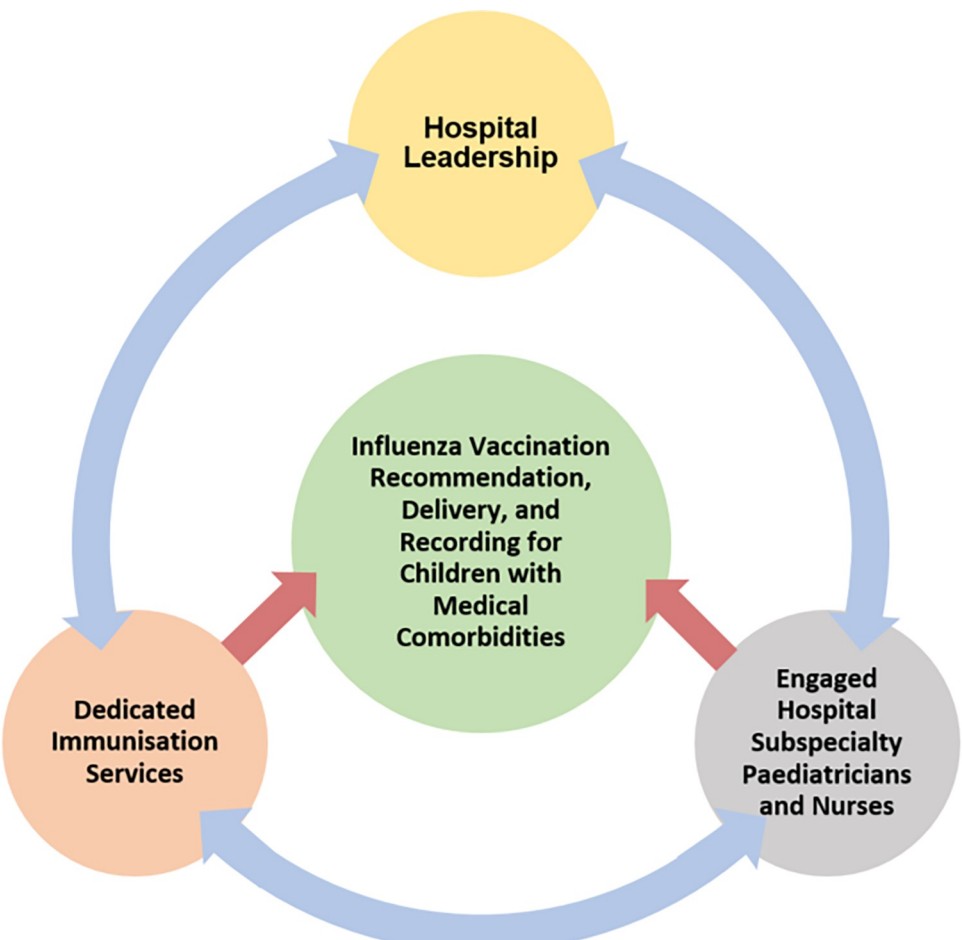

**Fig 1. The elements that directly and indirectly influence and drive influenza vaccination, recommendation delivery for children with comorbidities in Australian hospitals.**

due to the impact on their clinical workload. Although, hospital paediatricians have previously identified influenza disease to be a significant problem for children with comorbidities, influenza vaccination was a lower priority compared to other clinical problems [15]. Primary care physicians have previously identified vaccine recommendations for high-risk children should be given by their respective specialist paediatricians [16]. Given the perceptions strongly promoting hospital paediatricians' role in vaccine recommendation but not delivery, interventions and resources should facilitate easy hospital-based vaccine delivery.

Support from hospital leadership and having a dedicated immunisation service on site were identified by both paediatricians and nurses as key elements for successful vaccine delivery and recording for children with comorbidities. Hospital leadership was perceived as critical for creating a supportive hospital environment and culture for influenza vaccination delivery through financial support of dedicated immunisation services and prioritising vaccine promotion to clinical teams. Financial support and goal setting by hospitals have previously underpinned successful hospital staff influenza vaccination programs [17]. Likewise, dedicated immunisation services were seen to provide critical expert capacity for convenient, opportunistic hospital-based vaccine delivery and recording on the AIR for both influenza and NIP vaccines [8].

Dedicated immunisation services were identified by participating paediatricians to alleviate their concerns for increased clinical work and were perceived as the ideal providers for vaccine delivery. Participants from sites without dedicated immunisation services identified their vaccine delivery capacity to be strongly limited and would direct patients to outside vaccination services including hospitals with dedicated immunisation services. Introduction of temporary dedicated immunisation services for seasonal influenza vaccine delivery at Australian tertiary paediatric hospitals in 2010 has previously shown substantial improved uptake for children with comorbidities as well as capacity for providing cocooning vaccinations [18]. There is currently not funding of a specific influenza cocooning strategy in Australia. However, influenza vaccinations are currently funded nationally for children under 5 years old, individuals with comorbidities, adults over 65 years, indigenous peoples over 6 months old, and pregnant individuals [19]. Additionally, state-based funded influenza vaccination programs for all adults were introduced across Australia in 2022 [20]. Influenza vaccine delivery rose 226% across two paediatric hospitals with 420 and 949 doses given in 2009 and 2010 respectively. Clinicians support for and the presence of an on-site vaccination clinic has also previously increased influenza vaccine uptake in elderly patients, another high-risk group.

Other successful interventions for improving influenza vaccination in children include vaccine education [21], reminders [22] directed to parents or paediatricians and changes in the vaccine delivery process [23]. However, a review of interventions could not identify one superior intervention for increasing vaccine uptake [24]. Recently successful multi-component interventions, targeting providers, patients, and clinic barriers for vaccine uptake across a range of different comorbidity types, have been described [25–27]. Given the multifaceted and interlinked critical influences we found in this study, multi-component interventions may be needed to target multiple barriers simultaneously. Consequently, development of bespoke user-centred intervention 'bundles' are envisioned, guided by our current findings and pre-established frameworks including the *P3* model [28]. Accordingly, future reflection with paediatricians, nurses, hospital leadership, and patients is required for checking identified themes and evaluating specific interventions components and bundles.

This was the first multi-state study to examine clinician's attitudes for influenza vaccination in children with comorbidities. We captured a strong diversity of voices including paediatricians, immunisation leaders, and nurses from both paediatric and general hospital sites. However, we were limited by only being able to engage with hospital sites and a subset of the subspecialty teams that were willing to participate in the study. We did not engage with regional or rural hospital sites due to our capacity and resources. As such these results, may only be generalisable to the subspecialities interviewed at Australian speciality tertiary paediatric and general metropolitan hospitals, but not reflective of regional or rural hospitals nor subspecialities not interviewed. Potential barriers for influenza vaccination recommendation and delivery may have been missed due to our limited number of interviews and discussion groups. Additionally, we did not interview parents and caregivers of children with comorbidities, as such missed their perspectives on the barriers and facilitators for influenza vaccine recommendation and delivery.

In summary, Australian hospitals play a critical role in influenza vaccination of children with comorbidities. Supportive hospital leadership, dedicated immunisation services on site and engaged hospital paediatricians and nurses are the key elements for consistent influenza vaccine recommendations and delivery for children with comorbidities. Paediatricians were strongly influenced by belief in their role for recommending influenza vaccination, patients' comorbidities impacting influenza disease severity, and the impact of vaccination delivery on their clinical workload capacity. Furthermore, dedicated immunisation services and onsite vaccination resources were identified to be critical for vaccine delivery and recording. Support

by hospital leadership for hospital-based service delivery and dedicated immunisation services are paramount for their continuing success. Future interventions need to be informed by these key elements for increased likelihood of successful influenza vaccination programs in the hospital setting.

## Supporting information

**S1 Appendix. Clinical provider interview and discussion group session guide.**
(DOCX)

## Author Contributions

**Conceptualization:** Daniel A. Norman, Margie Danchin, Christopher C. Blyth, Hannah C. Moore, Holly Seale.

**Data curation:** Daniel A. Norman.

**Formal analysis:** Daniel A. Norman, Holly Seale.

**Funding acquisition:** Pamela Palasanthiran.

**Investigation:** Daniel A. Norman.

**Methodology:** Daniel A. Norman, Margie Danchin, Christopher C. Blyth, Holly Seale.

**Project administration:** Daniel A. Norman, Christopher C. Blyth, Pamela Palasanthiran, David Tran, Kristine K. Macartney, Ushma Wadia, Holly Seale.

**Resources:** David Tran, Kristine K. Macartney.

**Supervision:** Margie Danchin, Hannah C. Moore, Holly Seale.

**Validation:** Holly Seale.

**Visualization:** Daniel A. Norman.

**Writing – original draft:** Daniel A. Norman.

**Writing – review & editing:** Margie Danchin, Christopher C. Blyth, Pamela Palasanthiran, David Tran, Kristine K. Macartney, Ushma Wadia, Hannah C. Moore, Holly Seale.

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
