## [Decision Letter · Decision Letter 0]

30 Jun 2022

PONE-D-21-11166Australian hospital paediatricians and nurses’ perspectives and practices for influenza vaccine delivery in children with medical comorbiditiesPLOS ONE

Dear Dr. Norman,

Thank you for submitting your manuscript to PLOS ONE. After careful consideration, we feel that it has merit but does not fully meet PLOS ONE’s publication criteria as it currently stands. Therefore, we invite you to submit a revised version of the manuscript that addresses the points raised during the review process.

While I think this is an important topic that warrants investigation, there were some issues that are significant enough that they undermine the contributions of the study and prevent me from supporting its publication without any revision. Together with both reviewers I have a number of reservations about this paper. They are outlined below.

The funding for influenza vaccination in the cocoon strategy should be clarified.The study has some limitations to be taken into account and declared. Please, refer to the second reviewer’s comments point by point. Please submit your revised manuscript by August 14, 2022. If you will need more time than this to complete your revisions, please reply to this message or contact the journal office at plosone@plos.org. Please include the following items when submitting your revised manuscript:A rebuttal letter that responds to each point raised by the academic editor and reviewer(s). You should upload this letter as a separate file labeled 'Response to Reviewers'.A marked-up copy of your manuscript that highlights changes made to the original version. You should upload this as a separate file labeled 'Revised Manuscript with Track Changes'.An unmarked version of your revised paper without tracked changes. You should upload this as a separate file labeled 'Manuscript'.If applicable, we recommend that you deposit your laboratory protocols in protocols.io to enhance the reproducibility of your results. Protocols.io assigns your protocol its own identifier (DOI) so that it can be cited independently in the future. For instructions see: https://journals.plos.org/plosone/s/submission-guidelines#loc-laboratory-protocols. Additionally, PLOS ONE offers an option for publishing peer-reviewed Lab Protocol articles, which describe protocols hosted on protocols.io. Read more information on sharing protocols at https://plos.org/protocols?utm_medium=editorial-email&utm_source=authorletters&utm_campaign=protocols.

We look forward to receiving your revised manuscript.

Kind regards,

Prof. Maria Gańczak

Section Editor

PLOS ONE

Journal Requirements:

2. Please include in your Methods section (or in Supplementary Information files) the participating hospitals/institutions. Furthermore, please consider including more information on the number of interviewers, their training and characteristics.

4. Thank you for stating the following financial disclosure: "PP provided funding for DN's airline travel and accommodation during discussion groups and interviews conducted at Children’s Hospital at Westmead and Sydney Children’s Hospital, Randwick through her Clinical Research Funding from the Department of Immunology and Infectious Diseases, Sydney Children's Hospitals Network, Randwick. DN's airline travel and accommodation during discussions and interviews conducted at The Royal Children’s Hospital, Melbourne and Northern Hospital, Epping was provided through his University of Western Australia’s Medical School higher degree by research faculty funding. No other funding for this project was sought or provided.

The funders had no role in the study design, data collection and analysis, decision to publish, or preparation of the manuscript."

We note that one or more of the authors is affiliated with the funding organization, indicating the funder may have had some role in the design, data collection, analysis or preparation of your manuscript for publication; in other words, the funder played an indirect role through the participation of the co-authors. If the funding organization did not play a role in the study design, data collection and analysis, decision to publish, or preparation of the manuscript and only provided financial support in the form of authors' salaries and/or research materials, please do the following:

a. Review your statements relating to the author contributions, and ensure you have specifically and accurately indicated the role(s) that these authors had in your study. These amendments should be made in the online form.

b. Confirm in your cover letter that you agree with the following statement, and we will change the online submission form on your behalf: 

“The funder provided support in the form of salaries for authors [insert relevant initials], but did not have any additional role in the study design, data collection and analysis, decision to publish, or preparation of the manuscript. The specific roles of these authors are articulated in the ‘author contributions’ section.

5. Thank you for stating the following in the Competing Interests section: "I have read the journal's policy and the authors of this manuscript have the following competing interests: A/Prof Seale has previously received funding from vaccine manufactures for investigator driven research and for presenting at workshops. This funding was not associated with this research."

7. Your abstract cannot contain citations. Please only include citations in the body text of the manuscript, and ensure that they remain in ascending numerical order on first mention.

8. Your ethics statement should only appear in the Methods section of your manuscript. If your ethics statement is written in any section besides the Methods, please delete it from any other section. 

Reviewers' comments:

Reviewer's Responses to Questions

**Comments to the Author**

1. Is the manuscript technically sound, and do the data support the conclusions?

Reviewer #1: Yes

Reviewer #2: Yes

2. Has the statistical analysis been performed appropriately and rigorously? 

Reviewer #1: N/A

Reviewer #2: N/A

3. Have the authors made all data underlying the findings in their manuscript fully available?

Reviewer #1: Yes

Reviewer #2: Yes

4. Is the manuscript presented in an intelligible fashion and written in standard English?

Reviewer #1: Yes

Reviewer #2: Yes

5. Review Comments to the Author

Reviewer #1: What is not clear to the reader is the funding for influenza vaccination in the cocoon strategy. Do people in the cocoon pay for their vaccination, or does another institution cover the cost? Does vaccination coverage under the cocoon strategy depend on the diagnosed disease in the index person?

Reviewer #2: Dear Authors,

Thank you for your interesting article. It gives an interesting presentation on Australian situation on the topic. The study however has some limitations to be taken into account and declared.

Firstly, it included a small sample size which limits the transferability of results to other hospital settings. As you did not interview a clinician from all departments with HRC, you cannot rule out the possibility of missing additional themes or barriers.

It is not enough to interview only paediatricians and nurses to give vaccination recommendations against influenza for Australian children with comorbidities. While this study involved only a subset of HRC (High-risk children) and is limited by its small sample size, it suggests a strong need to engage parents in vaccine conversations. However, you were limited by only being able to engage with hospital sites and subspecialty teams willing to participate in the study; furthermore, participants in your study had generally positive attitudes towards the influenza vaccine. Health care workers who are highly disengaged with the value of influenza immunisation are unlikely to volunteer to be involved. Lastly, you did not undertake interviews with parents and so the comments around parental concerns about the influenza vaccination are based on the perceptions of the HCWs.

6. PLOS authors have the option to publish the peer review history of their article (what does this mean?). If published, this will include your full peer review and any attached files.

Reviewer #1: No

Reviewer #2: **Yes: **Chiara Cadeddu

---

## [Author Response · Author response to Decision Letter 0]

20 Sep 2022

20th of September 2022

Dear reviewers,

Thank you for your time in reviewing our submission. Please find below the details of our edits to the manuscript below and the new version of the manuscript uploaded.

We have clarified the funding of influenza vaccination in Australia and how this pertains to cocooning vaccination on lines 483 to 487 based on reviewer 1’s comments. Additionally, we have included further limitations as detailed by reviewer 2’s comments in lines 507 to 516. We have included further details about the interviews and discussion groups in the methods section in lines 139 to 153. We have modified the funding statement as per the publisher’s request on lines 36 to 47.

We have updated our reference list based on our edits from reviewers’ comments. These two new references include 

19. The Australian Technical Advisory Group on Immunisation. ATAGI advice on seasonal influenza vaccines in 2022: The Commonwealth of Australia; 2022 [cited 2022 10/08/2022]. Available from: https://www.health.gov.au/sites/default/files/documents/2022/02/atagi-advice-on-seasonal-influenza-vaccines-in-2022.pdf.

20. Queensland Government. Influenza (flu) vaccinations 2022: The State of Queensland; 2022. Available from: https://www.qld.gov.au/health/conditions/immunisation/free-influenza-vaccinations-2022. 

Daniel Norman, BSc, MInfDis, MPH PhD

Senior Research Officer, National Centre for Immunisation Research and Surveillance 

Corresponding author

Postal address: NCIRS, Kids Research, Sydney Children’s Hospitals Network, Cnr Hawkesbury Rd & Hainsworth St, Westmead Locked Bag 4001, Australia 

Telephone: (+61) 4229 9161

Email: da.norman@outlook.com

---

## [Editor Report · Decision Letter 1]

6 Nov 2022

Australian hospital paediatricians and nurses’ perspectives and practices for influenza vaccine delivery in children with medical comorbidities

PONE-D-21-11166R1

Dear Dr. Norman,

We’re pleased to inform you that your manuscript has been judged scientifically suitable for publication and will be formally accepted for publication once it meets all outstanding technical requirements.

Kind regards,

Maria Gańczak

Section Editor

PLOS ONE
---

## [Editor Report · Acceptance letter]

2 Dec 2022

PONE-D-21-11166R1 

Australian hospital paediatricians and nurses’ perspectives and practices for influenza vaccine delivery in children with medical comorbidities 

Dear Dr. Norman:

I'm pleased to inform you that your manuscript has been deemed suitable for publication in PLOS ONE. Congratulations! Your manuscript is now with our production department. 

Kind regards, 

on behalf of

Prof. Maria Gańczak 

Section Editor

PLOS ONE